# Utilizing Motion Capture Systems for Instrumenting the OCRA Index: A Study on Risk Classification for Upper Limb Work-Related Activities

**DOI:** 10.3390/s23177623

**Published:** 2023-09-02

**Authors:** Pablo Aqueveque, Guisella Peña, Manuel Gutiérrez, Britam Gómez, Enrique Germany, Gustavo Retamal, Paulina Ortega-Bastidas

**Affiliations:** 1Departamento de Ingeniería Eléctrica, Facultad de Ingeniería, Universidad de Concepción, Concepción 4070386, Chile; guisella.pena@biomedica.udec.cl (G.P.); enrique.germany@biomedica.udec.cl (E.G.); gustavo.retamal@biomedica.udec.cl (G.R.); 2Departamento de Ergonomía, Facultad de Ciencias Biológicas, Universidad de Concepción, Concepción 4070386, Chile; mangutie@udec.cl; 3Ingeniería Biomédica, Facultad de Ingeniería, Universidad de Santiago de Chile, Santiago 8320000, Chile; britam.gomez@usach.cl; 4Departamento de Kinesiología, Facultad de Medicina, Universidad de Concepción, Concepción 4030000, Chile; portegab@udec.cl; 5Programa de Doctorado en Ciencias de la Salud, Escuela Internacional de Doctorado, Universidad Rey Juan Carlos, 28943 Madrid, Spain

**Keywords:** musculoskeletal disorders, repetitive tasks, instrumented OCRA index, ergonomics

## Abstract

In the search to enhance ergonomic risk assessments for upper limb work-related activities, this study introduced and validated the efficiency of an inertial motion capture system, paired with a specialized platform that digitalized the OCRA index. Conducted in a semi-controlled environment, the proposed methodology was compared to traditional risk classification techniques using both inertial and optical motion capture systems. The inertial method encompassed 18 units in a Bluetooth Low Energy tree topology network for activity recording, subsequently analyzed for risk using the platform. Principal outcomes emphasized the optical system’s preeminence, aligning closely with the conventional technique. The optical system’s superiority was further evident in its alignment with the traditional method. Meanwhile, the inertial system followed closely, with an error margin of just ±0.098 compared to the optical system. Risk classification was consistent across all systems. The inertial system demonstrated strong performance metrics, achieving F1-scores of 0.97 and 1 for “risk” and “no risk” classifications, respectively. Its distinct advantage of portability was reinforced by participants’ feedback on its user-friendliness. The results highlight the inertial system’s potential, mirroring the precision of both traditional and optical methods and achieving a 65% reduction in risk assessment time. This advancement mitigates the need for intricate video setups, emphasizing its potential in ergonomic assessments.

## 1. Introduction

Musculoskeletal disorders (MSD) encompass injuries to muscles, nerves, tendons, joints, cartilage, and the lumbar spine [1,2]. These injuries arise from postural overload, repetitive movements, and muscle fatigue [3,4]. When they occur within a work setting, they are termed “Work-Related Musculoskeletal Disorders” (WRMSD), with the upper extremities being one of the most affected body areas [5,6,7]. These disorders predominantly arise in two scenarios:The work environment and job performance significantly contribute to the onset of the disorder. [8]Work conditions exacerbate or extend the duration of the injury. [8]

The economic ramifications of WRMSD are profound. For businesses, the consequences include increased absenteeism, reduced productivity, escalating healthcare costs, disability claims, and workers’ compensation [8]. As highlighted by the European Agency for Safety and Health at Work (EU-OSHA) in 2019, 60% of workers experiencing musculoskeletal issues linked them to their occupation. Echoing this, a 2014 National Health Survey in the US indicated that 11.2 million workers reported WRMSD, with the highest prevalence in the construction and public service sectors [9]. These statistics culminated in over 70 million medical visits and an average loss of 8 working days per affected individual, amounting to an annual cost of between 45 and 54 billion dollars [10].

Given the emphasis on occupational health and safety, ergonomics has taken center stage. It plays a pivotal role in fostering safer workplaces and assessing WRMSD risks. Implementing a comprehensive ergonomic strategy not only reduces the risk of occupational diseases and injuries, but also enhances worker well-being and mitigates company expenditures [11,12].

Ergonomics remains fundamental to occupational risk assessment, employing systematic strategies for risk identification and analysis [13,14,15]. These methodologies span from questionnaires to direct observations and precise measurements [11]. While traditional techniques, such as BORG [16], OWAS [17], RULA [18], REBA [19], OCRA [20], and NIOSH [21], provide insights into worker health and potential workplace hazards, their primarily retrospective and observational nature poses limitations [12]. Contemporary initiatives aim to digitalize these techniques, enabling centralized data storage and more precise recommendations [22,23]. However, face-to-face risk assessments remain the standard.

In recent times, advanced technological devices have been developed to record physiological and biomechanical data within work settings [24,25,26,27,28,29,30,31,32]. Among these, motion capture systems, both optical and inertial, are at the forefront. Optical systems demand extensive infrastructural modifications [30], whereas inertial systems, fitted with sensors like magnetometers and accelerometers, have gained traction due to their compact nature and adaptability [28,31,33].

While there are attempts to establish the use of sensors for monitoring MSD in the workplace, there are also challenges preventing their widespread adoption outside of academic settings [12]:Systems should not interfere with the worker’s tasks as they need to be applied in real environments, allowing for the acquisition of relevant data over extended periods.Concerns include data privacy and confidentiality, employee compliance, sensor durability, and potential cost–benefit outcomes for employers considering their use in the workplace.

Building upon these insights, this study employed the OCRA index method on the Unity^®^ platform to assess risks associated with workers’ upper limb repetitive activities, using an IMU-based system on an automated platform. The OCRA index (OI) considers factors such as rest and recovery times, force usage perception, high-risk postures, repetitive actions, and psychosocial elements, categorizing risks into “no risk”, “low risk”, and “risk” levels [18,34,35].

This study’s primary aim is to validate the efficacy of this novel approach in discerning an automated risk classification for upper limb work-related activities in a semi-controlled environment. Our proposal is contrasted against both traditional methodologies and an optical system, which serves as the benchmark for reference.

## 2. Development and Operation of Platform for WRSMD Determination

### 2.1. Upper Extremity Risk Assessment Method: OCRA Index

One method for evaluating the risk of upper extremity injuries in the workplace is the OCRA index [36,37]. The ISO 11228-3:2009 [38] standard, which outlines guidelines for the manual handling of low loads at high frequency, is used to calculate the recommended reference technical actions (RTA) for the evaluated work shift. The OCRA index value is obtained by dividing the actual technical actions (ATA) by the RTA, as shown in Equation (Equation 1). The OCRA index provides a numerical measure of the risk of upper extremity injuries associated with repetitive tasks in the workplace.
(1)$$\mathrm{OCRA}\;\mathrm{Index}=\frac{ATA}{RTA},$$

To obtain the number of actual technical actions (ATA), it is necessary to determine the number of technical actions ($n_{T}{}_{C}$) performed during work cycle times ($t_{C}$) in seconds. This information allows the calculation of frequency per minute (*f*) using Equation (Equation 2), which relates the number of technical actions performed to the duration of the work cycle. The frequency per minute is an important parameter for calculating the OCRA index, used to assess the risk of upper extremity injuries associated with repetitive tasks in the workplace.
(2)$$f=n_{T}{}_{C}\times \frac{60}{t_{C}},$$

Taking into consideration the net duration (*t*) of the repetitive task, the value of ATA can be obtained according to Equation (Equation 3)
(3)ATA=f×t,

On the other hand, to obtain the number of RTAs, the following must be considered:Frequency constant of technical actions per minute ($k_{f}$ = 30);Force multiplier, $F_{M}$, obtained from the perception of force use according to the Borg Scale or percentage of maximum voluntary contraction (%MVC);Posture multiplier, $P_{M}$, where awkward movements and postures at the shoulder, elbow, wrist, and hand are considered. A $P_{M}$ value is assigned to each of these parts of the arm and then the lowest value is assigned to the total task;Repetitiveness multiplier, $R_{e}{}_{M}$, where a ratio of repetitive motions to the duty cycle is obtained;Additional multiplier, $A_{M}$, which considers a series of observations of the psychosocial and organizational type of the work place;Multiplier recovery times, $R_{c}{}_{M}$, where work, rest, and lunch times during working hours are considered;Duration multiplier, $t_{M}$, where a factor is assigned according to the total time of repetitive work, in minutes;Net duration, in minutes, of repetitive work, *t*.

Finally, the RTA value is calculated with the Equation (Equation 4)
(4)$$RTA=k_{f}\times R_{c}{}_{M}\times t_{M}\times F_{M}\times P_{M}\times R_{e}{}_{M}\times A_{M}\times t,$$

Through comprehensive analyses, the OCRA index classifies the risk into three distinct levels: no risk, very low risk, and risk. The specific OI values delineating each risk level can be found in Table 1. While these values serve as our chosen thresholds, it is worth noting that they might undergo slight variations depending on updates to the OCRA methodology or its adaptations to particular national or sectoral nuances. The selection of these thresholds aligns with the latest guidelines for the application of the OI. These thresholds are rooted in in-depth epidemiological research that associates task repetitiveness and intensity with the prevalence of WRMSD among workers.

An example of evaluating a work task using the OCRA index for a worker who must assemble a part in 5 s is shown below. The worker picks up one piece with their right hand and the second piece with their left hand. Their workday lasts 480 min, without considering the 60 min of lunch, with 435 min of repetitive work (*t* = 435 and $t_{M}$ = 1). They have two breaks of 15 min each, one in the morning and the other in the afternoon during their workday. With this distribution, they work 5 h without adequate rest ($R_{c}{}_{M}$ = 0.45).

To analyze the work segment, two technical actions were identified: picking up the first component and positioning the component.

As the work cycle is 5 s, the frequency (*f*) of technical actions is 24 per minute. Using this, the ATA value can be obtained according to Equation (Equation 3). For this case, the ATA is 10.440.

Since this is an assembly task, the worker must apply force each time a component is assembled. Therefore, a force analysis is carried out by consulting the worker about their perception of the use of force according to the Borg Scale. In this case, a use of force score of 0.5 and 2 was determined for each technical action. With these values, the force multiplier value for the task was obtained, which corresponds to $F_{M}$ = 0.94.

To analyze risky postures, all the actions carried out during the work cycle are considered. In this case, the following were identified for the right upper extremity at the shoulder and hand level:Shoulder flexion/extension greater than 60° during 40% of the work cycle ($P_{M}$ = 1);Hand in pinch for more than 96% of the work cycle ($P_{M}$ = 0.6).

Finally, the $P_{M}$ value that represents the task is the one with the lowest score, $P_{M}$ = 0.6.

Since the work cycle considered for this example lasts 5 s and the tasks need to be executed for more than 50% of the cycle time, we obtain a value of $R_{e}{}_{M}$ = 0.7.

For this example, the presence of additional factors such as the use of individual protection equipment, blows in work stations, exposure to cold, vibrations, or inadequate work rhythms, was not considered, so AM = 1.

With all the factors obtained, a value of RTA = 2.318 can be calculated using Equation (Equation 4).

Therefore, an OCRA index value of 4.5 is obtained, which corresponds to a “Risk” classification level. Table 2 shows in detail all the values of interest considered in this example case.

### 2.2. Platform for Risk Assessment in the Upper Extremity

The developed platform in Unity^®^ (Unity Technologies, San Francisco, CA, USA), version 2020.3.14f1, incorporates the stages of risk analysis and classification.

For the analysis of repetitive actions in the upper extremities on the platform, a BVH file obtained from any motion capture platform is imported. These files provide the necessary data to represent movements in a biomechanical model of 61 segments. The model self-scales according to Chilean anthropometry when the height and sex of the person under evaluation are input [39,40]. Figure 1 and Figure 2 show the steps required to correctly represent the platform on the computer model, along with a screenshot of the platform.

To examine postures and movements captured by the movement systems, the platform offers the option of selecting the joint of interest, including the shoulder, elbow, wrist and hand. Once the area to be analyzed has been selected, the scroll bars can be used to select the areas of interest for capture. In this way, different actions can be studied independently. The platform automatically delivers the values of the angles during the entire selected time. Moreover, it allows for the entry of threshold angles (upper and lower) to identify risks. When the thresholds are set, the platform automatically delivers the time during which positions are adopted that exceed these limits. This procedure can be repeated for all joints of interest. Figure 3 shows the necessary steps to analyze joints of interest according to risk thresholds on the platform.

In terms of digitizing the OCRA index, the platform offers a comprehensive interface for carrying out all the necessary analyses. This involves following the flow shown in Figure 4, which includes the analysis stage of posture, force, repeatability, and additional factors.

Initially, the workday must be divided into hours, and the activities performed during each hour, including repetitive work, non-repetitive work, break, lunch, and end of day, must be entered. This will automatically calculate values such as *t*, $t_{M}$, and $R_{c_{M}}$. Subsequently, the technical actions complying with the ISO 11228-3:2009 [38] standard must be identified, specifying whether they correspond to the right or left arm, the number of repetitions and the duration in seconds. This will calculate $n_{T_{C}}$, *f*, and ATA. For the force analysis, the Borg or percent MVC scale should be selected to record the operator’s perceived force used in each technical action, determining the force score and the value of $F_{M}$. To analyze postures at shoulder, elbow, wrist, and hand level, the time in seconds that risk thresholds are exceeded must be input. A numerical $P_{M}$ is assigned to the elbow, wrist, and hand, and the lowest $P_{M}$ value is assigned to the subtask. For the shoulder, it only indicates if there is “risk” or “no risk”. Next, the repeatability multiplier is calculated by determining the percentage of time for each technical action. Finally, the psychosocial and organizational factors are considered, and the duration of its occurrence is entered. This will calculate $A_{M}$. Finally, the platform automatically delivers the RTA value and the corresponding OCRA index value with the risk classification corresponding to “risk”, “very low risk,” or “no risk,” which are represented by the colors green, yellow, and red, respectively.

### 2.3. Motion Capture Technologies

Motion capture systems are vital in biomechanical analysis. They predominantly fall into one of two categories: optical systems and inertial systems. In this study, both are used to assess risk and compare the performance of the proposed methodology.

#### 2.3.1. Optical System

This research utilizes the Optitrack^®^ system as the gold standard. It comprises 8 Prime*^X^* 22 cameras, each with a resolution of 22 MP, ±0.15 mm accuracy, and a native frame rate of 360 FPS. A biomechanic markerset of 39 markers was placed on a tight black suit worn by the subjects. The experiment was conducted in the Human Movement Laboratory of the Ergonomics Department at the Universidad de Concepción, as shown in Figure 5.

#### 2.3.2. Inertial System

A custom inertial motion capture system, composed of 18 measuring units communicating via Bluetooth 5.0 with a central acquisition computer, is also used. The sensors are organized in a tree topology network, where five of them function as central units and the others as peripherals. Each central unit is connected to specific peripheral sensors, and the data are recorded at a frequency of 100 Hz in frames of 8 bytes. A Bluetooth USB serial dongle is used to connect the five central units to the PC. The arrangement of the sensors on the body are shown in Figure 6 and has the following characteristics:Central Unit 1 is composed of the head sensor, which is connected to three peripheral sensors: thoracic spine, lumbar spine, and sacral spine.Central Unit 2 is composed of the right shoulder sensor, which is connected to three peripheral sensors: right arm, right forearm, and right hand.Central Unit 3 is composed of the left shoulder sensor, which is connected to three peripheral sensors: left arm, left forearm, and left hand.Central Unit 4 is composed of the right thigh sensor, which is connected to two peripheral sensors: right leg and right foot.Central Unit 5 is composed of the left thigh sensor, which is connected to two peripheral sensors: left leg and left foot.

### 2.4. Repetitive Action Test Evaluation Procedures

Twenty healthy subjects (10 females aged 27 ± 7 years, height 162 ± 4 cm; 10 males aged 28 ± 3 years, height 172 ± 7 cm) with no history of musculoskeletal injuries were recruited for this study. All participants provided written consent, and the protocol was approved by the Research and Development Ethics Committee of the Universidad de Concepción with code CEBB 794-2020.

For the determination of the number of participants, the sample size was selected based on the logistical feasibility, availability of participants matching the criteria, and considerations from previous similar studies that in most cases only report cases of study [30,31].

Each subject performed two repetitive action tests while wearing both the Optitrack suit and IMUs simultaneously. Additionally, video recordings were made using SONY^®^ cameras (Tokyo, Japan) model HDR-AS50 from frontal and lateral views.

The first test involved transferring four 0.5 kg weights from a shelf (139.5 cm high) to a bucket (77 cm high) and then placing the bucket on the floor. Figure 7 shows the sequence of actions performed by the subjects during test 1. The second test required subjects to assemble two pieces of a carcass (3.5 cm × 4.5 cm × 1.5 cm each) placed at different heights, then place the resulting carcass on a surface 61 cm high. Figure 8 shows the performance of the second test.

The technical actions identified in test 1 are taking weight, leaving weight in the bucket, picking up the bucket and leaving the package on the floor. On the other hand, the technical actions of test 2 are taking the upper part of the case, taking the lower part of the case, joining the case parts, and depositing the case in the shelf. Each action was repeated four times in both tests.

For both tests carried out, the following considerations for the working day were taken into account:A repetitive work time of 450 min was considered ($t_{M}$ = 1);Three rest periods of 10 min each were considered, leaving 3 h of the workday without adequate rest. ($R_{c}{}_{M}$ = 0.7);None of the technical actions performed in the tests exceeded 50% of the duty cycle time, nor were the duty cycles less than 15 s ($R_{e}{}_{M}$ = 1);The presence of any additional factors during the workday was not considered ($A_{M}$ = 1);The Borg Scale was used to obtain $F_{M}$. Each of the subjects was asked their perception of use of force according to this scale.

### 2.5. Analysis of Postures and Joint Ranges for Risk Classification

Video recordings were analyzed using Kinovea^®^ version 0.9.5 software [41]. The angle tool was used to identify postures and joint ranges that exceeded risk thresholds for all areas of the arm. The OCRA index was manually determined based on the collected data.

BVH files were obtained from motion capture methods, and a comparison between the video camera image and the representation in the developed platform is shown in Figure 9. The same procedure was used for the optical and inertial system. Data files were imported into the Unity platform, and analysis tools were used to identify technical actions performed in each test. The risk thresholds specified in the ISO standard were entered for each joint in the upper extremity and the time (in seconds) that these thresholds were exceeded was obtained. The posture data were then used to calculate the OCRA index value and risk classification in an automated manner using the digitized OCRA index method.

## 3. Results

First, the shoulder, elbow, and wrist movements were analyzed. Then, times in which risky postures were adopted for the elbow and wrist were compared since they are the areas in which a numerical value is assigned to the PM factor.

Statistical analyses were conducted with a confidence level of 95%. The Shapiro–Wilk [42] test was applied to assess the normality of the data groups, where variables with a *p*-value > 0.05 provided sufficient evidence to consider a normal distribution. Subsequently, the homogeneity of variances of the data was analyzed, confirming that the variables with a *p*-value > 0.05 complied with the homogeneity assumption. To determine whether there are significant differences between the data, parametric and nonparametric ANOVA tests [43,44] were applied according to the distribution of the data. In case that the data distribution was not normal, the Welch ANOVA was applied [45]. The results are presented in Table 3.

In the case of elbow flexion/extension, significant differences were observed between test 1 and test 2 for the right segment. Consequently, the Tukey Multiple Comparisons Test was applied, revealing that test 1 exhibited a significant difference in the optical system. This indicates that the optical system provides different values compared to the inertial and traditional systems. For test 2 in the right segment, the traditional method showed variations in relation to the values obtained from the inertial and optical systems. Regarding elbow pronation/supination, a comparison was only made between the optical and inertial systems, as this movement could not be captured by the video cameras. The analysis of the tests revealed that both the optical and inertial systems produced similar results, with no significant variations observed within this range. For wrist flexion/extension, significant differences were found in all the tests, suggesting that it cannot be concluded that the three methods yield similar values for this posture.

After analyzing the risk times, the $P_{M}$ values were obtained. Table 4 displays the $P_{M}$ values for test 1 and test 2 using the three analysis methods.

Test 1 results indicated that, for the shoulder, all subjects, except subject 9, were rated “risky” due to lifting weights from the highest surface of the ledge. For the elbow, values of $P_{M}$ were mostly identical, with slight differences in motion capture systems due to access to all joint movements. For the wrist, differences were observed between the optical system and the inertial and traditional systems, due to the occlusion of markers on the hand during weight deposition. The traditional method consistently yielded a $P_{M}$ value of 1 for all cases.

For test 2, the highest risk values were obtained in the right shoulder, elbow, and wrist due to the pinch grip to take the first piece of carcass from the top surface of the shelf, which resulted in more pronounced movements for shorter subjects (<170 cm). Lower values of $P_{M}$ were obtained for the left segment, and a greater number of coincidences was observed between the three analysis methods in the elbow and wrist. The optical system showed lower values of $P_{M}$ for the wrist in some cases due to occlusions during the assembly of the case.

After conducting the posture analysis, the $P_{M}$ value for the general task was determined for each test, considering the one with the lowest value. The $F_{M}$ values were also obtained for each subject based on their perception of the use of force according to the Borg Scale. With all the factors obtained, the OCRA index value was finally calculated for each test and subject. The final results of $F_{M}$, $P_{M}$, and OI for test 1 and test 2 for the right and left segments are shown in Table 5.

The obtained results of $F_{M}$ indicate that force was used to carry out test 1. Differences between subjects were due to their sex and physical condition. Regarding the final $P_{M}$ values for test 1, these indicate that risky postures and movements were adopted since all subjects obtained a final value of less than 1. The OCRA index values for this test classify it as risky. All subjects obtained the same risk classification except for subjects 1 and 19, who were affected by a change in posture value due to the inertial system.

For test 2, the value of $F_{M}$ obtained for both segments is 1, indicating minimal effort to assemble the cases for the subjects. The final value of $P_{M}$ for the right segment indicates risky postures and movements when taking a piece of the carcass from the upper part of the shelf, while the value of $P_{M}$ for the left segment is 1 for all subjects, except when markers were occluded with the optical system. The OCRA index value for both segments indicates a non-risky test, with higher values in the right segment due to arm elevation.

The study compared the OCRA index values obtained using different methods, with the optical system considered as the gold standard. The error between the optical and inertial system was found to be ±0.098, and the error between the optical system and the traditional method was ±0.137. These small error values suggest that both the traditional method and the inertial system provide accurate OCRA index values, and the risk classification for the evaluations performed remains unchanged. A statistical analysis was performed using the OCRA index values to determine if there are significant differences among the values obtained by different systems. The results are shown in Table 6. Firstly, the Shapiro–Wilk test was applied to determine if the data follow a normal distribution with a confidence level of 95%. Then, the results were compared, leading to the conclusion that there are no significant differences in the data among the systems, and they have the same mean and median. In other words, similar results are obtained regardless of the system used.

In addition, with the risk classification results for all measurement systems, a comparison was obtained from both instrumented alternatives (optical and inertial) with respect to the traditional approach, and metrics of accuracy, precision, sensitivity, and F1-score were obtained for each risk level. These values are shown in Table 7.

The performance metrics obtained from the comparison indicate that the inertial system is capable of correctly identifying the “risk” and “no risk” classifications with an F1-score value of 0.97 and 1, respectively. However, for the “very low risk” classification, there is a value of 0.67, which, despite being higher than 0.5, indicates that this situation is not always adequately identified. In contrast, the metrics obtained from the optical method indicate that it is always able to correctly identify the three risk categories, with F1-score values of 1 each.

To compare the analysis time and time classification, the duration of the evaluation with OCRA index was recorded from the motion capture files and the video recordings obtained by the traditional method. Table 8 shows the times separated by test and by system.

With respect to the evaluation times, a 65% reduction in the duration of risk analysis and classification was obtained with motion capture technologies. This is due to the fact that, with the use of the platform, the process of identifying technical actions and risk postures is faster and simpler due to the objective information provided by the systems used.

On the other hand, one of the main advantages of motion capture systems based on inertial sensors is their portability and suitability for work environments. Although the system used in this research might seem complex in its arrangement, it is essential in order to assess the user’s perception in terms of comfort. All participants in this study correctly followed calibration protocols and reported no discomfort while using the sensorized system during the execution of their tasks. This feedback is particularly important because, even though the system comprises multiple measuring units, its design allowed participants to continue with their work activities unhindered.

## 4. Discussion

This article tested a platform for assessing ergonomic risks in the upper extremities and demonstrated that the use of an inertial system and the digitization of evaluation methods can expedite the risk classification process by providing real-time information.

The statistical analysis conducted on the duration of risky postures (Table 3) revealed significant differences in values among the optical, inertial, and traditional systems. This indicates that each method provides distinct values. The optical system is affected by occlusions that occur during the tests when performing flexion/extension movements and leaving the weight inside the bucket in test 1. On the other hand, the traditional evaluation only relied on video cameras from a frontal and lateral view, which did not allow for a comprehensive analysis of all adopted postures. Therefore, a separate analysis was conducted for elbow pronosupination, comparing only the optical and inertial systems, which yielded similar values. A comparison with the traditional method was not possible since it is typically impractical to position a specific video camera for the movement in workplace settings. When comparing the posture factors (see Table 4), this information is corroborated as the factor values are more consistent between each other in the elbow.

The OCRA index results (Table 5 and Table 6) indicate no significant difference between the evaluation methods, suggesting similar indices obtained with all three applied methods. However, the inertial system exhibits an error of ±0.098 compared to the optical system, which does not represent a significant change in risk classification. Moreover, the analysis of risk classification (see Table 7) indicates that the inertial system accurately classifies risk levels as “risk” and “no risk” with F1 scores of 0.97 and 1, respectively.

Finally, the proposed methodology achieved a 65% reduction in the analysis and risk classification times (Table 8). The platform streamlines and simplifies the process of identifying technical actions and risky postures through the objective information provided by the utilized systems.

Regarding the complications that arose during the development of this research, there was the constant calibration that must be performed to the optical systems. In the case of the application of the traditional method, the evaluation turned out to be more complicated due to the arrangement of the video cameras used, since the subjects were wearing the motion capture system suits and some movements could not be correctly identified. In addition, some fast wrist movements did not allow their correct analysis by the video software.

On the other hand, future work related to this research consists of increasing the number of subjects and testing in work environments.

The inertial system with the proposed analysis methodology is an effective tool for risk assessments in occupational environments due to its compact size and portability. It provides objective information on risky postures during repetitive actions and enables the establishment of risk thresholds, expediting the calculation of posture factors. This eliminates the constraint of video camera availability in the workplace and removes visual aspects dependent on subjective interpretation by evaluators, which vary based on individual experience.

## 5. Conclusions

The utilization of motion capture systems, both optical and inertial, offers a significant advantage in instrumenting the OCRA index for risk classification related to upper limb work activities. Our findings validate the efficacy of these systems in a semi-controlled environment, with the optical system emerging as the gold standard. Both methods were found to provide accurate OCRA index values when compared to traditional methods, with negligible error margins. Importantly, the inertial system showcased its potential in real work environments due to its portability and comfort, with participants reporting no discomfort. Additionally, motion capture technologies allowed for a 65% reduction in risk analysis duration, streamlining the evaluation process. While the optical system consistently achieved high accuracy across all risk categories, the inertial system showed areas for improvement, particularly in the “very low risk” classification. Overall, our study underscores the potential of motion capture technologies in advancing risk classification methodologies, combining efficiency, accuracy, and user comfort.

## Figures and Tables

**Figure 1 sensors-23-07623-f001:**
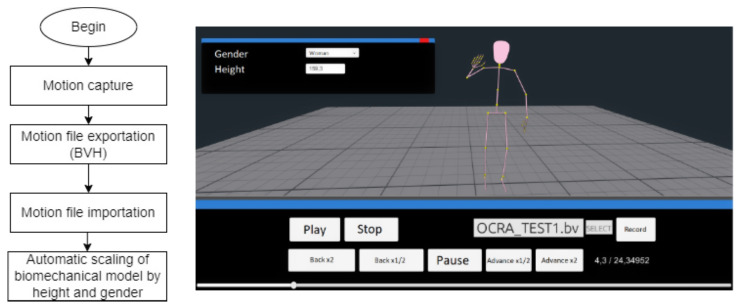
Methodological diagram of the importation of the BVH file and adjustment of the computer dummy in the platform for risk determination.

**Figure 2 sensors-23-07623-f002:**
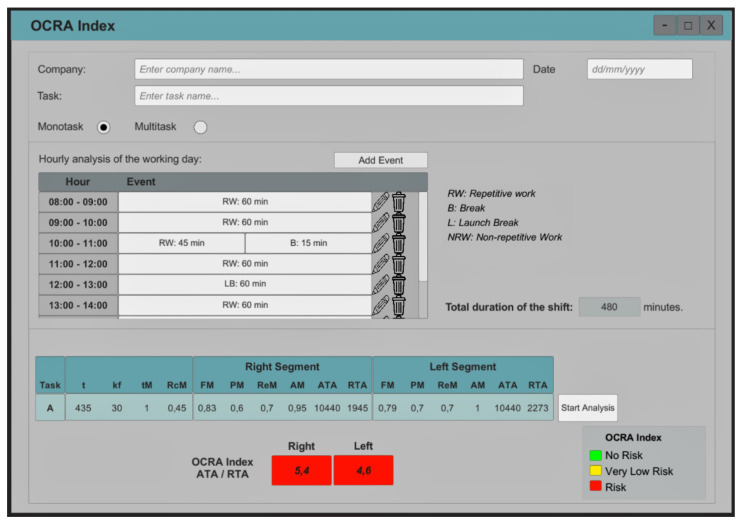
Screenshot of the platform window with finished OCRA index evaluation.

**Figure 3 sensors-23-07623-f003:**
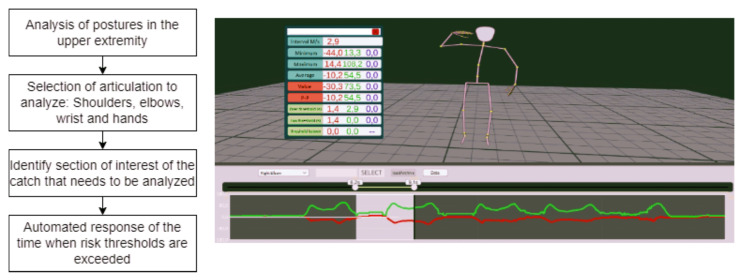
Schematic for selecting the area of interest to perform joint posture analysis according to risk thresholds. The green line corresponds to elbow flexion/extension and the red line corresponds to elbow pronation/supination.

**Figure 4 sensors-23-07623-f004:**
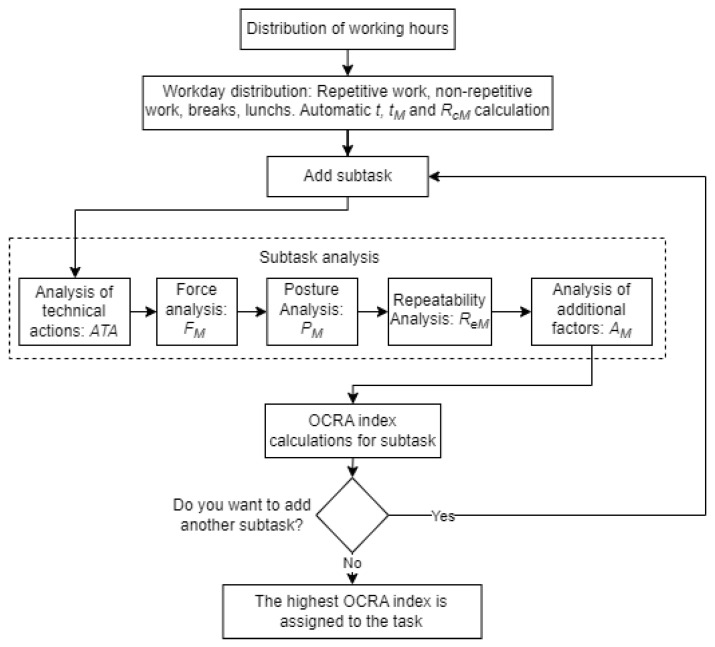
Diagram for entering information into the digitized OCRA index platform.

**Figure 5 sensors-23-07623-f005:**
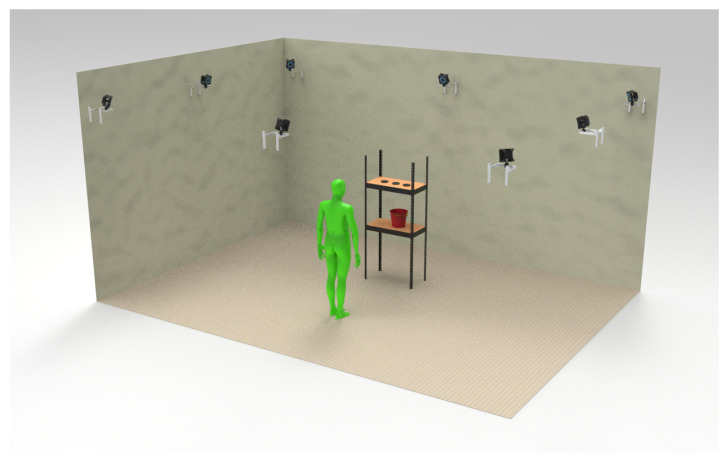
Motion laboratory consisting of optoelectronic system and evaluation elements: height adjustable shelf, bucket and 0.5 kg weights.

**Figure 6 sensors-23-07623-f006:**
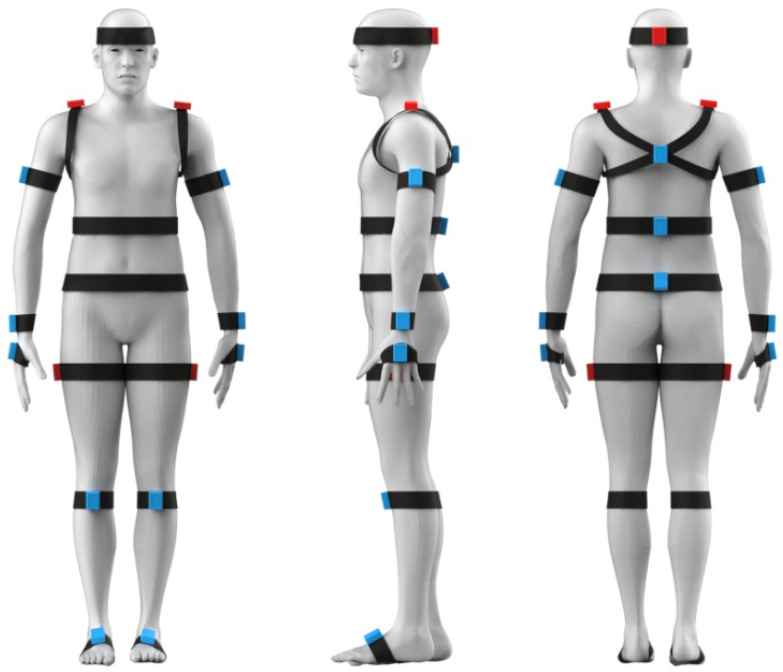
Arrangement of inertial sensors. The red sensors represent the central units and the blue sensors are the peripheral units.

**Figure 7 sensors-23-07623-f007:**
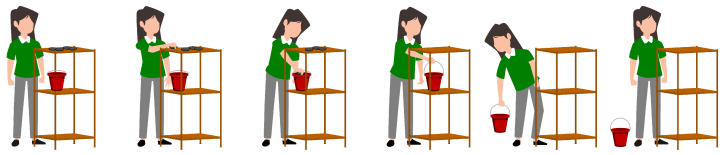
Sequence of actions performed by volunteers for test 1.

**Figure 8 sensors-23-07623-f008:**
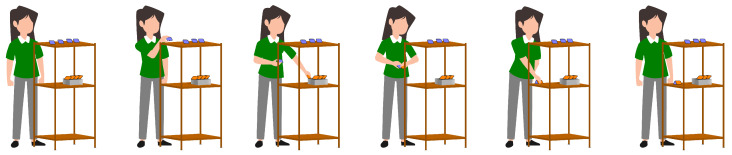
Sequence of actions performed by volunteers for test 2.

**Figure 9 sensors-23-07623-f009:**
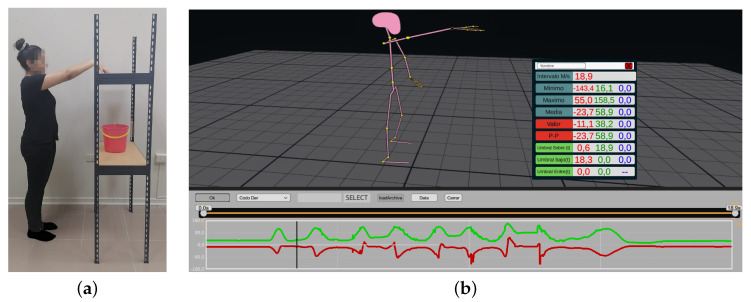
Comparison of image captured with video camera and mannequin representation on the developed platform. (**a**) Video recording capture for test 1. (**b**) Capture of platform for test 1.

**Table 1 sensors-23-07623-t001:** Risk classification according to OCRA Index.

OCRA Index Value	Risk Classification
OI < 2.3	No risk
2.3 ≤ OI ≤ 3.5	Very low risk
3.5 > OI	Risk

**Table 2 sensors-23-07623-t002:** Evaluation results with OCRA Index for example case.

Factors	Value
Repetitive work net time, *t*. In minutes.	435
Work cycle time, $t_{C}$. In seconds.	5
Number of technical actions per cycle, $n_{T}{}_{C}$	2
Frequency of technical actions, *f*	24/per minute
Constant of frequency, $k_{f}$	30
Duration multiplier, $t_{M}$	1
Recovery multiplier, $R_{c}{}_{M}$	0.45
Force multiplier, $F_{M}$	0.94
Posture multiplier, $P_{M}$	0.6
Repetitiveness multiplier, $R_{e}{}_{M}$	0.7
Additional multiplier, $A_{M}$	1
ATA value	10.440
RTA value	2.318
**OCRA index**	**4.5**
**Risk classification**	**Risk**

**Table 3 sensors-23-07623-t003:** Statistical analysis of elbow and wrist risk posture maintenance times. Tests 1 and 2.

Risk Posture	Test	System	Data Distribution	Homogeneity of Variances	Comparison of Medians
**Elbow flexo/extension**	Test 1. Right Segment	Optical	Normal *	No	Significant difference
		Inertial	Not normal		
		Traditional	Not normal		
	Test 2. Right Segment	Optical	Normal *	Yes	Significant difference
		Inertial	Not normal		
		Traditional	Normal *		
	Test 2. Left Segment	Optical	Not normal	Yes	No significant difference *
		Inertial	Normal *		
		Traditional	Normal *		
**Elbow prone/supination**	Test 1. Right Segment	Optical	Not normal	Yes	No significant difference *
		Inertial	Not normal		
	Test 2. Right Segment	Optical	Normal *	Yes	No significant difference *
		Inertial	Not normal		
	Test 2. Left Segment	Optical	Normal *	Yes	No significant difference *
		Inertial	Not normal		
**Wrist flexo/extension**	Test 1. Right Segment	Optical	Normal *	No	Significant difference
		Inertial	Not normal		
		Traditional	Not normal		
	Test 2. Right Segment	Optical	Normal *	Yes	Significant difference
		Inertial	Normal *		
		Traditional	Normal *		
	Test 2. Left Segment	Optical	Normal *	No	Significant difference
		Inertial	Not normal		
		Traditional	Not normal		

* Denotes significance *p*-value > 0.05.

**Table 4 sensors-23-07623-t004:** Results of $P_{M}$ in shoulder, elbow, and wrist for test 1 and test 2. Right and left segments.

Subject	Method	$P_{M}$ for Test 1. Right Segment			$P_{M}$ for Test 2. Right Segment			$P_{M}$ for Test 2. Left Segment		
		Shoulder	Elbow	Wrist	Shoulder	Elbow	Wrist	Shoulder	Elbow	Wrist
**1**	Optical	Risk	0.7	0.7	No Risk	0.7	1	No Risk	0.7	1
	Inertial	Risk	1	0.6	No Risk	0.7	1	No Risk	0.7	1
	Traditional	Risk	1	1	No Risk	0.7	0.7	No Risk	1	1
**2**	Optical	Risk	0.7	1	No Risk	0.7	1	No Risk	1	1
	Inertial	Risk	1	1	No Risk	0.7	1	No Risk	1	1
	Traditional	Risk	1	1	No Risk	1	0.7	No Risk	1	1
**3**	Optical	Risk	1	1	Risk	1	1	No Risk	1	1
	Inertial	Risk	1	0.7	Risk	1	1	No Risk	1	1
	Traditional	Risk	1	1	Risk	1	1	No Risk	1	1
**4**	Optical	Risk	1	0.7	No Risk	0.7	0.7	No Risk	1	1
	Inertial	Risk	1	1	No Risk	0.7	1	No Risk	0.7	1
	Traditional	Risk	1	1	No Risk	0.7	0.7	No Risk	1	1
**5**	Optical	Risk	1	1	Risk	1	1	No Risk	1	1
	Inertial	Risk	0.7	0.7	Risk	1	1	No Risk	1	1
	Traditional	Risk	1	1	Risk	1	0.7	No Risk	1	1
**6**	Optical	Risk	1	0.7	Risk	1	1	No Risk	1	1
	Inertial	Risk	1	1	Risk	1	1	No Risk	1	1
	Traditional	Risk	1	1	Risk	1	0.7	No Risk	1	1
**7**	Optical	Risk	1	0.7	Risk	0.7	0.7	No Risk	1	1
	Inertial	Risk	1	1	Risk	0.7	1	No Risk	1	1
	Traditional	Risk	1	1	Risk	1	0.7	No Risk	1	1
**8**	Optical	Risk	1	1	Risk	1	1	No Risk	1	1
	Inertial	Risk	1	0.6	Risk	1	0.7	No Risk	1	1
	Traditional	Risk	1	1	Risk	1	0.7	No Risk	1	1
**9**	Optical	No Risk	0.7	1	No Risk	1	1	No Risk	1	1
	Inertial	No Risk	0.7	0.6	No Risk	1	0.7	No Risk	1	1
	Traditional	No Risk	0.7	1	No Risk	1	0.7	No Risk	1	1
**10**	Optical	Risk	0.7	1	No Risk	1	1	No Risk	1	1
	Inertial	Risk	1	1	No Risk	0.7	0.7	No Risk	1	1
	Traditional	Risk	1	1	No Risk	1	0.7	No Risk	1	1
**11**	Optical	Risk	1	0.7	Risk	0.7	0.7	No Risk	1	0.7
	Inertial	Risk	0.7	0.7	Risk	0.7	1	No Risk	1	1
	Traditional	Risk	1	0.7	Risk	0.7	0.7	No Risk	1	1
**12**	Optical	Risk	0.7	0.7	No Risk	0.7	0.6	No Risk	1	0.6
	Inertial	Risk	0.7	1	No Risk	1	1	No Risk	1	1
	Traditional	Risk	0.7	1	No Risk	1	0.7	No Risk	1	1
**13**	Optical	Risk	0.6	0.7	Risk	0.7	0.6	No Risk	1	1
	Inertial	Risk	0.7	1	Risk	0.7	1	No Risk	1	1
	Traditional	Risk	0.7	1	Risk	0.7	0.7	No Risk	1	1
**14**	Optical	Risk	0.7	0.6	Risk	1	1	No Risk	1	1
	Inertial	Risk	1	1	Risk	1	1	No Risk	1	1
	Traditional	Risk	1	1	Risk	0.7	0.7	No Risk	1	1
**15**	Optical	Risk	0.7	1	No Risk	0.7	1	No Risk	1	1
	Inertial	Risk	1	1	No Risk	1	1	No Risk	1	1
	Traditional	Risk	1	1	No Risk	0.7	0.7	No Risk	1	1
**16**	Optical	Risk	1	0.7	Risk	0.7	1	No Risk	1	0.7
	Inertial	Risk	1	1	Risk	1	1	No Risk	1	1
	Traditional	Risk	1	1	Risk	1	0.7	No Risk	1	1
**17**	Optical	Risk	0.7	0.7	Risk	0.7	0.7	No Risk	1	1
	Inertial	Risk	0.6	1	Risk	0.7	0.7	No Risk	1	1
	Traditional	Risk	0.7	1	Risk	1	0.7	No Risk	1	1
**18**	Optical	Risk	0.7	0.6	Risk	0.7	0.7	No Risk	1	0.7
	Inertial	Risk	1	0.6	Risk	1	1	No Risk	1	1
	Traditional	Risk	1	0.6	Risk	1	0.7	No Risk	1	1
**19**	Optical	Risk	1	0.7	Risk	0.7	0.7	No Risk	1	0.7
	Inertial	Risk	0.7	0.6	Risk	1	1	No Risk	1	1
	Traditional	Risk	1	1	Risk	1	0.7	No Risk	1	1
**20**	Optical	Risk	0.7	0.6	Risk	0.7	0.7	No Risk	0.7	1
	Inertial	Risk	1	1	Risk	0.7	0.7	No Risk	1	1
	Traditional	Risk	1	1	Risk	1	0.7	No Risk	1	1

*P_M_*: Posture multiplier. Values are dimensionless.

**Table 5 sensors-23-07623-t005:** Results of $F_{M}$, $P_{M}$, and OI for test 1 and test 2. Right and left segments.

Subject	System/Method	Test 1. Right Segment			Test 2. Right Segment			Test 2. Left Segment		
		$F_{M}$	$P_{M}$	**OI**	$F_{M}$	$P_{M}$	**OI**	$F_{M}$	$P_{M}$	**OI**
**1**	Optical	0.59	0.7	3.2	1	0.7	1.5	1	0.7	1.0
	Inertial	0.59	0.6	3.8	1	0.7	1.5	1	0.7	1.0
	Traditional	0.59	0.7	3.2	1	0.7	1.5	1	1	1.0
**2**	Optical	0.56	0.7	4.6	1	0.7	2.2	1	1	1.1
	Inertial	0.56	0.7	4.6	1	0.7	2.2	1	1	1.1
	Traditional	0.56	0.7	4.6	1	0.7	2.2	1	1	1.1
**3**	Optical	0.59	0.7	6.8	1	1	1.4	1	1	1.0
	Inertial	0.59	0.7	6.8	1	1	1.4	1	1	1.0
	Traditional	0.59	0.7	6.8	1	1	1.4	1	1	1.0
**4**	Optical	0.59	0.7	3.9	1	0.7	1.6	1	1	0.7
	Inertial	0.59	0.7	3.9	1	0.7	1.6	1	0.7	1.0
	Traditional	0.59	0.7	3.9	1	0.7	1.6	1	1	0.7
**5**	Optical	0.56	0.7	3.9	1	0.7	2.0	1	1	0.9
	Inertial	0.56	0.7	3.9	1	0.7	2.0	1	1	0.9
	Traditional	0.56	0.7	3.9	1	0.7	2.0	1	1	0.9
**6**	Optical	0.59	0.7	4.2	1	0.7	2.2	1	1	1.1
	Inertial	0.59	0.7	4.2	1	0.7	2.2	1	1	1.1
	Traditional	0.59	0.7	4.2	1	0.7	2.2	1	1	1.1
**7**	Optical	0.59	0.7	4.0	1	0.7	1.6	1	1	0.7
	Inertial	0.59	0.7	4.0	1	0.7	1.6	1	1	0.7
	Traditional	0.59	0.7	4.0	1	0.7	1.6	1	1	0.7
**8**	Optical	0.59	0.7	3.9	1	0.7	1.8	1	1	0.8
	Inertial	0.59	0.6	4.6	1	0.7	1.8	1	1	0.8
	Traditional	0.59	0.7	3.9	1	0.7	1.8	1	1	0.8
**9**	Optical	0.59	0.7	4.2	1	0.7	1.5	1	1	0.7
	Inertial	0.59	0.6	4.8	1	0.7	1.5	1	1	0.7
	Traditional	0.59	0.7	4.2	1	0.7	1.5	1	1	0.7
**10**	Optical	0.56	0.7	3.9	1	0.7	1.6	1	1	0.7
	Inertial	0.56	0.7	3.9	1	0.7	1.6	1	1	0.7
	Traditional	0.56	0.7	3.9	1	0.7	1.6	1	1	0.7
**11**	Optical	0.59	0.7	3.6	1	0.7	1.6	1	0.7	1.1
	Inertial	0.59	0.7	3.6	1	0.7	1.6	1	1	0.8
	Traditional	0.59	0.7	3.6	1	0.7	1.6	1	1	0.8
**12**	Optical	0.65	0.6	3.9	1	0.6	1.9	1	0.6	1.3
	Inertial	0.65	0.6	3.9	1	0.7	1.6	1	1	0.8
	Traditional	0.65	0.6	3.9	1	0.7	1.6	1	1	0.8
**13**	Optical	0.65	0.6	7.3	1	0.6	2.5	1	1	1.0
	Inertial	0.65	0.7	6.3	1	0.7	2.1	1	1	1.0
	Traditional	0.65	0.7	6.3	1	0.7	2.1	1	1	1.0
**14**	Optical	0.59	0.6	9.0	1	0.7	2.1	1	1	1.0
	Inertial	0.59	0.7	7.7	1	0.7	2.1	1	1	1.0
	Traditional	0.59	0.7	7.7	1	0.7	2.1	1	1	1.0
**15**	Optical	0.59	0.6	3.4	1	0.7	1.2	1	1	0.5
	Inertial	0.59	0.6	3.4	1	0.7	1.2	1	1	0.5
	Traditional	0.59	0.6	3.4	1	0.7	1.2	1	1	0.5
**16**	Optical	0.59	0.7	3.7	1	0.7	1.5	1	0.7	1.0
	Inertial	0.59	0.7	3.7	1	0.7	1.5	1	1	0.7
	Traditional	0.59	0.7	3.7	1	0.7	1.5	1	1	0.7
**17**	Optical	0.59	0.7	4.2	1	0.7	2.1	1	1	1.0
	Inertial	0.59	0.6	4.8	1	0.7	2.1	1	1	1.0
	Traditional	0.59	0.7	4.2	1	0.7	2.1	1	1	1.0
**18**	Optical	0.59	0.6	4.0	1	0.7	2.1	1	0.7	1.4
	Inertial	0.59	0.6	4.0	1	0.7	2.1	1	1	1.0
	Traditional	0.59	0.6	4.0	1	0.7	2.1	1	1	1.0
**19**	Optical	0.56	0.7	3.5	1	0.7	1.4	1	0.7	1.0
	Inertial	0.56	0.6	4.1	1	0.7	1.4	1	1	0.7
	Traditional	0.56	0.7	3.5	1	0.7	1.4	1	1	1.7
**20**	Optical	0.62	0.6	3.2	1	0.7	1.4	1	0.7	0.9
	Inertial	0.62	0.7	2.8	1	0.7	1.4	1	1	0.7
	Traditional	0.62	0.7	2.8	1	0.7	1.4	1	1	0.7

*F_M_*: Force multiplier, *P_M_*: Posture multiplier, OI: OCRA index. All values are dimensionless.

**Table 6 sensors-23-07623-t006:** Statistical analysis of OCRA index values. Test 1 and 2.

Test	System	Data Distribution
**Test 1. Right Segment**	Optical	Not normal
	Inertial	Not normal
	Traditional	Not normal
**Test 2. Right Segment**	Optical	Normal *
	Inertial	Not normal
	Traditional	Not normal
**Test 2. Left Segment**	Optical	Normal *
	Inertial	Not normal
	Traditional	Not normal

* Denotes significance *p*-value > 0.05.

**Table 7 sensors-23-07623-t007:** Performance from instrumented OCRA index risk classification methods (optical and inertial) compared with the traditional system.

Methods	Metrics	Risk	Very Low Risk	No Risk
**Inertial**	Sensibility	0.95	0.5	1
	Precision	1	1	1
	Accuracy	0.97	0.97	1
	F1-Score	0.97	0.67	1
**Optical**	Sensibility	1	1	1
	Precision	1	1	1
	Accuracy	1	1	1
	F1-Score	1	1	1

**Table 8 sensors-23-07623-t008:** Risk classification times by traditional evaluation method vs. optical and inertial system.

Subject	Analysis and Classification Time Test 1			Analysis and Classification Time Test 2		
	Traditional [mm:ss]	Optical [mm:ss]	Inertial [mm:ss]	Traditional [mm:ss]	Optical [mm:ss]	Inertial [mm:ss]
1	33:12	10:52	10:10	37:10	15:21	14:58
2	33:56	09:57	09:30	36:24	14:01	13:40
3	32:45	09:34	09:25	37:20	15:10	14:45
4	34:02	10:12	09:13	36:34	15:11	15:10
5	32:01	11:12	10:57	37:09	13:53	13:41
6	34:20	11:20	11:05	38:10	16:30	15:52
7	34:33	10:22	09:45	38:58	13:36	12:06
8	33:58	09:13	08:59	40:37	14:58	14:58
9	32:20	12:35	11:54	39:22	15:41	16:10
10	33:34	09:34	09:20	37:58	14:57	14:57
11	33:59	09:53	09:41	37:47	15:31	16:12
12	33:30	10:08	09:54	38:23	15:23	15:17
13	32:19	11:43	11:12	36:34	14:45	14:32
14	34:28	11:34	11:29	37:19	15:41	15:23
15	33:26	10:50	10:08	38:02	15:10	15:07
16	35:30	09:58	09:40	40:42	15:30	15:25
17	34:40	10:40	10:33	39:45	14:12	14:10
18	33:58	11:50	11:24	37:58	14:57	14:31
19	34:13	12:35	12:10	38:50	13:33	13:25
20	33:15	11:57	11:32	37:49	15:24	15:14
**Average**	**33:30**	**10:36**	**10:30**	**38:32**	**15:31**	**14:29**

## Data Availability

Data are unavailable due to privacy and ethical restrictions.

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
