# Peer review of "Utilizing Motion Capture Systems for Instrumenting the OCRA Index: A Study on Risk Classification for Upper Limb Work-Related Activities"

_sensors, 2023, doi:10.3390/s23177623_

Round 1

Reviewer 1 Report

Dear authors,

I think you have a good, interesting article, which should be better highlighted. I recommend:

a) review the "Introduction" section (it should be better argued, from the articles already published in the field, in the international literature). You have few submissions, especially as such evaluations with Motion Capture Systems are many, on different aspects, including the one promoted by you

b) the conclusions...actually there are no conclusions in your text, there are discussions and results. In other words, the central ideas must be systematized and formulated as conclusions. You have written far too much in this section. The conclusions must be short, include the essence of the study and bring an overall picture, not technical details. Please review this section. Much of what you wrote in "Conclusions" should be moved to the "Discussions" section. By the way, don't you have this section in your work? shouldn't it, according to the instructions in the authors' guide????

c) the bibliographic references are few, especially for such an article with such a theme. Maybe it would be advisable to see other sources

d) revise the expression, to make it easier to understand for the readers of a newspaper. Your article is indeed a specialist article, but this does not mean that it cannot be transposed in a form that allows the information to be understood by a non-specialist. There is a lack of overall understanding of the ideas, because the language used in the article is difficult, cumbersome if you do not pay very, very close attention. This aspect could make your article not be studied correctly and carefully.

I hope my tips help you. Certainly, the other reviewers will also give you other advice, and the editor will decide.

Success

Author Response

Please kindly check the attachment.

Reviewer 2 Report

The authors address a quite interesting topic. However, some flaws are present in their work that need to be considered.

1)     The Abstract needs to be rewritten; it does not provide sufficient information on the methodology, the results, and the usefulness of the proposed work.

2)     References seem quite outdated; a related work section should be added with recent bibliography in the field of risk assessment with digital tools (inertial, optical, marker/markerless..). Lines 38-44 cannot be considered sufficient to frame the context of these technologies in the assessment of postural risks.

3)     Section 2.1: what is ATR (lines 69, 85)? If it is an acronym, it must be specified explicitly the first time it appears. It seems the authors are referring to RTA.

4)     Lines 93-94: why a break of 15 minutes “at the end of the workday”? It doesn’t seem to make sense.

5)     Line 114: authors should give details on the factors not considered.

6)     Lines 135-136: how are the threshold selected? It seems they are taken from the OCRA index, but the authors must be clearer.

7)     Figure 4: an “e” is missing. Authors must revise the manuscript to correct typos and English language (e.g., lines 164-166).

8)     Section 2.3.1: the description of the adopted motion capture technologies is detailed, but quite confusing concerning the methodology followed by the authors.

9)     Results:

a.      did the authors compute the number of participants necessary to conduct a significant statistical analysis (e.g., power analysis)? How was the number of participants determined?

b.      Table 3: how were data with non-normal distributions analysed?

c.      Table 7: results are not clearly reported. The authors collect results typical of a classification, but they indicate these as results from a comparison between two assessing methodologies; to compare performances of two methods in terms of classification metrics, it seems reasonable to report metrics for both classifications. If the inertial and the traditional were used as ground truth, it must be specified; there cannot be two different ground truth for the same labelling. Moreover, did the authors use a classifier?

10)  The IMUs arrangement seems quite cumbersome; the authors should take this aspect into account when comparing their approach with optical and traditional ones. Particularly, authors state that the inertial system is an effective tool in occupational environments due to its compact size and portability. Was the comfort of the participant assessed in any way with reference to this system?    

English language must be revised.
